# Intra Uterine Insemination in Two Couples with HPV Detection by Hyaluronidase-Based Swim-up Washing: Cases Report

**DOI:** 10.3390/jpm13010006

**Published:** 2022-12-20

**Authors:** Ilaria Cosci, Luca De Toni, Francesca Vasoin De Prosperi, Carrie Bedoni, Rafael Ramirez, Grazia Buonfantino, Alberto Ferlin, Andrea Garolla

**Affiliations:** 1Veneto Institute of Oncology IOV—IRCCS, 35128 Padova, Italy; 2Unit of Andrology and Reproductive Medicine, Department of Medicine, University of Padova, 35128 Padova, Italy; 3Garolla Clinic—IUI Center, 35128 Padova, Italy

**Keywords:** virology, pregnancy, biochemistry

## Abstract

Human papillomavirus (HPV) infection is the most common sexually transmitted disease worldwide and is associated with negative reproductive outcomes because of which it is recommended to postpone medically assisted reproduction (MAR). This raises major concerns for elder infertile couples. We showed that a hyaluronidase-based sperm washing (IALu) procedure blunted the HPV viral load in semen. Here, were report two cases of couples with long-term idiopathic infertility, ascribed to persistent semen HPV detection, finding a beneficial outcome from the use of IALu protocol applied to intra-uterine insemination (IUI). Case 1: A Caucasian couple (female aged 32, male aged 35), complained of having been attempting pregnancy for 4 years. HPV-DNA (genotypes 51 and 54) was detected on sperms. After a first unsuccessful IUI cycle attempt, using standard swim-up selection of spermatozoa, a second IUI cycle using the IALu procedure was associated with a pregnancy and a successful trimester of gestation. Case 2: A Caucasian couple (female aged 43, male aged 52) complained of having been attempting pregnancy for 3 years and showed the detection of HPV-DNA (genotype 66) on sperms. After a first unsuccessful standard IUI cycle attempt, two further IUI cycles using IALu procedure were pursued. The last cycle was associated with a pregnancy and a successful trimester of gestation. Although preliminary, the IALu procedure is a promising approach for straightforward fertility treatments in cases of recurrent HPV-DNA semen detection, avoiding critical latencies.

## 1. Introduction

Human papillomavirus (HPV) infection is the most common sexually transmitted disease worldwide, being detectable in 75% of sexually active people [1]. HPV is the causative agent of benign diseases, such as genital warts, but is also a recognized oncogenic virus, distinguished into high-risk and low-risk genotypes according to the oncogenic potential [2]. In addition to the aforementioned HPV-related diseases, an increasing number of evidence supports a role of HPV detection in subjects with poor fertility outcome. The prevalence of semen HPV detection greatly varies among available studies, from 0% in teenagers who had never had intercourse, to 100% in men with intrameatal warts [3]. In general, semen detection in asymptomatic subjects ranges from 2% to more than 30% according to patient characteristics such as age, lifestyles, sexual orientation, and no less important, the eventual partner positivity [4,5]. The pattern of HPV genotypes detected in semen is also widely variable, with HPV 6/11, 16, 18, 31, or 33 among the most frequent genotypes [6]. In particular, high-risk carcinogenic types can be detected in up to 66.7% of HPV-positive individuals [4]. The detection of HPV in semen has been associated with an increased risk of infertility of nearly three times compared to the negative population [7]. Accordingly, HPV-DNA-positive semen has been associated with altered sperm motility parameters and development of anti-sperm antibodies [8]. Negative reproductive outcomes following natural or assisted reproduction, such as up to six-fold higher risk of spontaneous abortions [9], has been mainly associated with the detection of HPV-DNA on sperm surface (reviewed in [8,10]). The acknowledgment of all this evidence by the European Society of Human Reproduction and Embryology (ESHRE) has been formalized in 2021 through the release of the guidelines for medically assisted reproduction (MAR) in patients with viral infection or disease [11]. ESHRE guidelines recognize that HPV detection in semen is the only viral factor having a clinical association with MAR outcomes although accounting for the challenging diffusion of the HPV-DNA test and the need of further widenings in the field of medically assisted reproduction. Accordingly, the opportunity of an HPV testing in semen should discussed with couples undergoing IUI, reminding couples with a known positive HPV test that HPV is a transient infection, and the postponing of MAR treatment is an option depending on the individual circumstances [11]. On these bases, outpatient counseling represents a key step in the clinical management of an infertile couple undergoing MAR procedures and having HPV detection in semen as the major microbiological factor [12]. Currently, the range of options available to minimize the lag time to HPV negativization includes the adoption of safe sex intercourse and personal hygiene practice to avoid recurrences [13]. On the other hand, the use of adjuvant vaccination, although not curative, showed a significant shortening of the time to semen negativization of HPV detection [14]. However, both cases involve the postponing of MAR treatment from several months to even years, a critical condition when elder couples face the option of fertility seeking [15]. Since the detection of HPV on sperms may have an impact on sperm quality and reproductive outcomes [16], the opportunity to reduce the semen viral load of HPV in order to improve the reproductive outcome in couples with idiopathic infertility associated with HPV detection has recently been addressed by our group. Whilst on one hand, standard washing procedures involving sperm washing medium, density gradient isolation, or swim-up selection are inadequate to reduce the seminal viral load, on the other hand, a virtual elimination of HPV adhering to the sperm surface was obtained through the application of a hyaluronidase-based sperm washing (IALu) procedure [17,18]. Here, were report a series of two cases of couples with long-term idiopathic infertility, both having an history of repeated MAR failures and persistent semen HPV detection as major negative microbiological factor, that agreed to undergo intra-uterine insemination (IUI) carried out with the application of an IALu procedure.

## 2. Cases Report

The study was approved by the Institutional Ethics Committee of the University Hospital of Padova, Italy, by the protocol no. 2336 and subsequent amendments. All participants had provided informed consent.

### 2.1. Case 1

In July 2020, a Caucasian couple was referred to our center for idiopathic infertility, seeking natural parenthood by non-protected sexual intercourse since March 2016. Both partners had normal karyotype and genetic screening for the *CFTR* gene [19].

The female partner was 32 years old, nulliparous with negative clinical history of major gynecological problems, and had a BMI of 25 kg/m^2^. The menstrual cycle was a regular 28 days, the estimation of ovarian reserve showed serum levels of anti-Müllerian hormone of 2.7 pmol/L and an antral follicle count of 14. Transvaginal ultrasound and tubal patency were normal. HPV and Papanicolaou stain (PAP) tests were negative.

The male partner was 35 years old, with negative andrological history. Semen analysis according to WHO 2010 [20] showed normozoospermia with detection of anti-sperm antibodies (56%) at spermMar test kit for IgG and IgA (FertiPro NV, Beernem, Belgium). Semen microbiological evaluation, including mycoplasma and chlamydia, was negative. INNO-LiPA Genotyping Extra II assay (INNO-LiPA; Fujirebio Europe, Ghent, Belgium), based on the principle of reverse hybridization after highly sensitive PCR amplification with SPF10 primers [21], was used for screening and genotyping for HPV-DNA in semen and reported the detection of genotypes 51 and 54. Cytological HPV localization by fluorescence in situ hybridization for HPV-DNA (HPV-FISH) [22] showed positive detection of HPV-DNA in 33% of sperms and 8% of exfoliated cells.

After outpatient counseling, based on normal sperm parameters and the female’s age, the couple was suggested to undergo IUI under stimulated cycles [23]. In a first cycle, the female partner received 100 IU/day of recombinant follicle stimulating hormone (FSH, Puregon, MSD Italia S.r.l.) for three days since the third day of menstrual cycle, followed by 75 IU/day FSH from the 6th to 11th day. Ovulation was induced at the 12th day with 6500 IU of human chorionic gonadotropin (hCG, Ovitrelle, Merck Serono, Copenhagen, Denmark) when three follicles had a diameter >17 mm, and the leading follicle had 19 mm diameter. The IUI was performed 36 h later, upon confirmation of the ovulation by the three major follicle, using standard partner’s semen preparation. Briefly, after 3 days of ejaculatory abstinence, the semen sample obtained by masturbation was allowed to liquefy for 30 min at 37 °C in a sterile container. Sperm cells were then washed with sperm washing medium (SWM, FUJIFILM Europe B.V., Tilburg, The Netherlands) and underwent swim-up selection under standard conditions at 37 °C with SWM as selection medium [18]. Sperm parameters and HPV-DNA test results at basal and after selection are summarized in Table 1. After IUI, the luteal phase was supported by 90 mg of vaginal progesterone supplementation until 14 days after insemination, when β-hCG assay was performed. This first treatment had an unsuccessful result.

A second IUI cycle attempt was performed the following month. The female partner underwent the same protocol of stimulation, induction of ovulation, and follow-up. At ovulation induction, four follicles had a diameter >17 mm, with two leading follicles having a diameter of 20 mm. Differently from the first attempt, the semen sample underwent preparation according to the IALu protocol. The application of the IALu protocol consisted of a first washing of liquified semen with SWM, followed by centrifugation at 1000× *g* for 10 min. The pellet of spermatozoa was then resuspended and incubated with hyaluronidase (FertiCult Flushing medium, FertiPro, Beernem, Belgium) at the concentration of 80 IU/mL at 37 °C for 30 min. The excess reagent was removed by standard washing with SWM followed by centrifugation. Swim-up selection of motile spermatozoa then was performed as described above. An aliquot of residual selected sperm sample was tested for HPV-DNA genotyping and HPV-FISH. Sperm parameters and results of HPV-DNA test at basal and after selection are summarized in Table 1. The achievement of pregnancy was tested by β-hCG assay showing a positive result. At the time of the drafting of this report, the achieved pregnancy had a normal first trimester of gestation as monitored through transvaginal ultrasound, showing a morphologically normal fetus with physiological heart-rate.

### 2.2. Case 2

In April 2021, a Caucasian couple was referred to our center for idiopathic infertility, seeking natural parenthood by non-protected sexual intercourse since January 2020. Both partners had normal karyotype and screening for the *CFTR* gene.

The female partner was 43 years old, nulliparous with negative clinical history for major gynecological issues, and had a BMI of 23 kg/m^2^. The menstrual cycle was a regular 27 days, the estimation of ovarian reserve showed serum anti-Müllerian hormone levels of 1.6 pmol/L and an antral follicle-count of 7. Transvaginal ultrasound and tubal patency were normal. HPV and PAP tests were negative.

The male partner was 52 years old, with negative andrological history. Semen analysis showed normozoospermia according to WHO 2010, with negative detection of anti-sperm antibodies. Semen microbiological evaluation, including mycoplasma and chlamydia, showed negative results. HPV-DNA genotyping reported the detection of genotype 66, with the HPV-FISH analysis showing positive detection of HPV-DNA in 23% of sperms and 14% of exfoliated cells.

Despite the elder age of the couple and the long-term infertility, a second-level MAR was suggested, and patients denied such approach, opting for the IUI treatment upon stimulated cycles.

In a first cycle, the female partner received 100 IU/day of purified urinary FSH (Meropur, Ferring B.V., Milan, Italy) for three days since the third day of the menstrual cycle, followed by 75 IU/day of FSH from the 6th to 11th day. Ovulation was induced at the 12th day with 6500 IU of hCG, when two follicles had a diameter >17 mm, and the leading follicle had 20 mm diameter. The IUI was performed 36 h later, upon confirmation of the ovulation, using a standard preparation of the semen sample through swim-up selection. Sperm parameters and results of HPV-DNA assessment at basal and after preparation are summarized in Table 1. After IUI, the luteal phase was supported by 90 mg of vaginal progesterone supplementation, until 14 days after insemination when the β-hCG assay was performed. The first treatment had an unsuccessful result.

A second IUI cycle attempt was performed the following month. The female partner received 150 UI/day of purified urinary FSH (Meropur) for three days since the third day of the menstrual cycle, followed by 100 IU/day from the 6th to 11th day. Ovulation was induced at the 12th day with 6500 IU of β-hCG, when two follicles had a diameter >17 mm, and the leading follicle had 20 mm diameter. Differently from the first attempt, the modified semen sample preparation by IALu-protocol was applied. An aliquot of residual selected sperm sample was tested for HPV-DNA genotyping and HPV-FISH. Sperm parameters and results of HPV-DNA assessment at basal and after preparation are summarized in Table 1. This attempt also had an unsuccessful result.

The following month, a third IUI cycle attempt was performed as the previous, with the sole exception of a stimulation with purified urinary FSH 150 UI/day (Meropur) from the 3rd day of menstrual cycle to the 11th day. Ovulation was induced at the 12th day with 6500 IU of β-hCG, when two follicles had a diameter >17 mm, and the leading follicle had 20 mm diameter. The IALu-protocol was applied for semen preparation. An aliquot of residual selected sperm sample was assessed for HPV-DNA genotyping and HPV-FISH. Sperm parameters and results of HPV-DNA assessment at basal and after preparation are summarized in Table 1. The achievement of pregnancy was tested by β-hCG assay, showing a positive result. At the time of the drafting of this report, the achieved pregnancy had a normal first trimester of gestation as monitored through transvaginal ultrasound, showing a morphologically normal fetus with physiological heart-rate.

## 3. Discussion

This is the first report on the application of modified swim-up protocol involving hyaluronidase treatment for semen sample preparation in two long-term idiopathic infertile couples with positive detection of HPV-DNA in semen. 

The role of HPV presence in semen as a cause of male/couple infertility is a controversial topic. Patients with HPV-DNA detection in semen often present as asymptomatic or with variable alterations of the seminal parameters, mostly related to cell motility [24,25]. Despite these inconsistencies, there is a relatively shared view according to which positive detection of HPV in semen is associated with increased risk of miscarriage and reduced chance of ongoing pregnancy [16]. Accordingly, the 2021 ESHRE guidelines for MAR in couples with HPV semen detection suggest discussing the delay of the fertility seeking in order to overcome reproductive issues [11], which involves major concerns due the estimated average time to clear HPV DNA detection [12]. Considering the prevalence of HPV genital detection in infertile patients, ranging from 12.5 to 20%, and the rising female age at first MAR attempt [26], postponing the access to MAR techniques is a critical point. Importantly, the detection of HPV DNA bound to the sperm surface requires addressing [27,28]. As previously suggested for HIV infection [29], the reduction of HPV viral load in semen may represent a straightforward approach to improve the reproductive outcome in these couples. To this regard, the tenacity with which HPV binds to the sperm surface has been a matter of concern since the first reports in the late 1990s, in which Brossfield et al. recognized the inability to completely remove the viral load from sperms by washing methods such as centrifuge, two-layer isolating colloidal wash, or yolk buffer procedures [30]. Interestingly, a very recent paper from Fenizia et al. showed that swim-up selection of spermatozoa succeeded in blunting HPV-DNA detection in 15 clinically HPV-positive subjects [31]. On the other hand, in the mid-2010s, we reported that the preliminary application of a Heparinase III-based sperm wash showed promising results in terms of viral load reduction [13]. However, this technique was not applicable since Heparinase III in not approved for the use in MARs.

Importantly, the rational for the use of Heparinase III was the ability of the hydrolytic enzyme to recognize a particular carbohydrate moiety of sydecanes, the sperm glycoprotein to which HPV is believed to bind [32]. Thus, heparinase was expected remove the surface binding sites of HPV to the spermatozoa with minimal involvement of cellular function. It is interesting to note that Heparinase III-sensitive carbohydrate moiety of sydecanes are structurally similar to those displayed by hyaluronic acid [33], on which the hydrolytic enzyme hyaluronidase is active. In addition to the common recognition of syndecan by most of HPV genotypes, a major advantage of the use of hyaluronidase is its clinical approval for oocyte denudation [34], making it a readily usable device for the clinical reduction of HPV viral load in semen even in cases of detection of multiple genotypes. To this regard, we recently showed a virtual elimination of HPV adhering to the sperm surface through the application of a hyaluronidase-based sperm washing procedure, practically independent of the HPV genotype and here re-named as the IALu protocol [18]. In the present study, we observed that in both cases, the adoption of the IALu protocol was associated with the decrease in the total motile count while undergoing IUI. Although it is an aspect that deserves further research, it seems to be linked to the sperm quality of the fresh sample. In both cases, in fact, sperm motility in the fresh sample at the first attempt was higher than in subsequent attempts.

We can surely account for some drawbacks in the present study: first and foremost is the observational and preliminary nature that two cases reports entail. Indeed, we described two cases reporting, respectively, four years and one year of unsuccessful unprotected sexual intercourse. At this level, the detection of HPV in semen has an indicative and not a causative role regarding the idiopathic infertility found in the participants. In addition, we recognize some “selection bias” since the reports included only HPV-positive cases, excluding the negative ones. However, the very preliminary use of the IALu protocol shows promising results essentially for its clinical safety, encouraging much larger prospective studies. To this regard, we previously showed that the in vitro application of the IALu protocol was not associated with major DNA sperm alterations [18]. In particular, in our opinion, it is important to underline some possible cues of the IALu protocol: (i) it can be readily performed in any MAR lab environment; (ii) it is intended to shorten the latency associated with natural virus clearance; (iii) it can be performed from the lowest level of assisted reproduction to a single-cell approach [18]; and (iv) relying on sperm processing, it finds no age restriction. Assuming that these preliminary results require a thorough clinical validation, we support the interest in using the IALu protocol in couples showing long-term idiopathic infertility associated with HPV semen detection, including those cases showing increased levels of anti-sperm antibodies with no signs of lower urinary tract syndrome or negative microbiological evaluation of semen [35].

## 4. Conclusions

This is the first report of two currently ongoing pregnancies obtained after the application of a modified swim-up sperm selection protocol involving hyaluronidase. Although preliminary, it might represent a new approach for the straightforward research of fertility in cases of recurrent HPV-DNA semen detection, shortening latencies in cases of persistent infection and/or critical aging. To accomplish this prospect, further clinical trials aimed to address the pregnancy rate associated with sperm preparation using the IALu protocol compared to recommended preparation methods such as swim-up technique and density gradient isolation are warranted.

## Figures and Tables

**Table 1 jpm-13-00006-t001:** Sperm parameters and details of human papillomavirus DNA detection in the male subjects at both basal conditions before semen preparation and upon sperm preparation by swim-up or IALu protocol.

	Basal Sperm Parameters before Preparation							Sperm Parameters after Preparation		
	Total Sperm Count (10^6^ Cells/Ejaculate)	Motility A+B (%)	Viability (%)	Normal Morphology (%)	ASA (%)	HPV-DNA Genotyping	HPV-FISH-Positive Cells	Total Motile Sperms (10^6^ Cells)	HPV-DNA Genotyping	HPV-FISH (% Positive Cells)
Case 1										
Attempt 1 Swim-up	151.0	40	78	20	56	51, 54	33% sperms 8% exfoliated	16	51, 54	16% sperms
Attempt 2 Swim-up + IALu	152.4	39	75	14	56	51, 54	24% sperms 6% exfoliated	15	n.d.	n.d.
Case 2										
Attempt 1 Swim-up	323.0	68	78	6	n.d.	66	23% sperms 14% exfoliated	50.4	66	14% sperms
Attempt 2 Swim-up + IALu	258.0	48	67	14	n.d.	66	24% sperms 12% exfoliated	24.4	n.d	n.d.
Attempt 3 Swim-up + IALu	331.2	59	77	20	n.d.	66	18% sperms 9% exfoliated	26.4	n.d.	n.d.

Abbreviations: HPV, human papillomavirus; ASA, anti-sperm antibodies; FISH, fluorescent in situ hybridization; IALu, hyaluronidase-based sperm washing procedure; n.d., non detectable.

## Data Availability

Data will be made available upon request.

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
