# Peer review of "Intra Uterine Insemination in Two Couples with HPV Detection by Hyaluronidase-Based Swim-up Washing: Cases Report"

_jpm, 2022, doi:10.3390/jpm13010006_

Round 1

Author Response

There are a number of comments to the work that require additions:

  1. Section 1. Introduction should be revised and supplemented with information on the incidence of different HPV genotypes, including in sperm, and their association with delayed fertility, and the development of reproductive problems in male individuals.

Answer

In agreement with the Reviewer’s request, we improved the introduction section by adding that the prevalence of semen HPV detection greatly varies among available studies, from 0% in teenagers who had never had intercourse, to 100% in men with intrameatal warts. In general, semen detection in asymptomatic subjects ranges from 2% to more than 30% ac-cording to patients characterisics, such as age, life styles, sexual orientation and, not less important, the eventual partner positivity. The pattern of HPV genotypes detected in semen is also widely variable, with HPV 6/11, 16, 18, 31 or 33 are among the most frequent genotypes. In particular, high-risk carcinogenic types can be detected in up to 66.7% of HPV positive individuals. The detection of HPV in semen has been associated with an increased risk of infertility of nearly 3 times, compared to the negative population. Accordingly, HPV-DNA positive semen has been associated with altered sperm motility pa-rameters, development of anti-sperm antibodies. Negative reproductive outcomes following natural or assisted reproduction, such as up to six-fold higher risk of spontaneous abortions, has been mainly associated with the detection of HPV-DNA on sperm surface.

  1. Section 4. Conclusion be supplemented with information on the prospects for further clinical research and development on the use of the modified IALu protocol in couples showing long-term idiopathic infertility associated with the detection of HPV in semen.

Answer

In agreement with the Reviewer’s suggestion, we clarified that, in order to accomplish the prospects of our preliminaru data, further clinical trials aimed to address the pregnancy rate associated with sperm preparation using of IALu protocol, compared to recommended preparation methods such as swim-up technique and density gradient isolation, are warranted.

Reviewer 2 Report

Article is about novel strategy for helping infertile couples with hpv. Several comments need to be addressed

Abstract should be shortened by brief description of conclusions in both reported cases.

Specific meaning of sentences needs double checking (e.g - resulted into a successful trimester of gestation, ww look forward etc )

A comment about Total motile sperms decreasing after procedure should be inserted.

A more specific description about hyaluronidase treatment and in what other domain could be used would add more value to the manuscript

Author Response

Reviewer 2

Article is about novel strategy for helping infertile couples with hpv. Several comments need to be addressed

1- Abstract should be shortened by brief description of conclusions in both reported cases.

Answer

We streamlined the cases description in the abstract as requested.

2- Specific meaning of sentences needs double checking (e.g - resulted into a successful trimester of gestation, ww look forward etc )

Answer

We clarified the sentences as requested

3- A comment about Total motile sperms decreasing after procedure should be inserted.

Answer

In agreement with the Reviewer’s request, we commented that in both cases the adoption of the IALu protocol was associated with the reduction in the total motile count undergoing IUI. Although it is an aspect that deserves further widenings, it seems to be linked to the sperm quality of the fresh sample. In both cases, in fact, sperm motility in the fresh sample at the first attempt was higher than in subsequent attempts.

3- A more specific description about hyaluronidase treatment and in what other domain could be used would add more value to the manuscript.

Answer

In agreement with the Reviewer’s suggestion, we thoroughly described the IALu procedure and suggested its use also in those cases showing increased levels of anti-sperm antibodies with no signs of lower urinary tract syndrome or negative microbiological evaluation of semen.

Reviewer 3 Report

Intra Uterine Insemination in Two Couples with HPV Detection, by Hyaluronidase-based Swim-up Washing: Cases Report

-          Very good structure of the article

-          Well explained technique

-          Add information about sperm motility improvement if found in order to sustain literature data cited in introduction section

-          Further research needed in order prove the utility of Swim-up technique in daily practice for HPV positive patients

Author Response

Reviewer 3

Intra Uterine Insemination in Two Couples with HPV Detection, by Hyaluronidase-based Swim-up Washing: Cases Report

- Very good structure of the article

- Well explained technique

Answer

We thank the Reviewer for his/her positive comments

- Add information about sperm motility improvement if found in order to sustain literature data cited in introduction section

- Further research needed in order prove the utility of Swim-up technique in daily practice for HPV positive patients

Answer

As for point 2 of Reviewer 1 and point 3 of Reviewer 2, we commented motility variations upon IALu application and the research needed in order prove the utility of IALu technique in daily practice for HPV positive patients.